# Improving the Pervaporation Performance of PDMS Membranes for Trichloroethylene by Incorporating Silane-Modified ZSM-5 Zeolite

**DOI:** 10.3390/polym15183777

**Published:** 2023-09-15

**Authors:** Xiaosan Song, Xichen Song, Yue Zhang, Jishuo Fan

**Affiliations:** 1School of Environmental and Municipal Engineering, Lanzhou Jiaotong University, Lanzhou 730070, China; 12211118@stu.lzjtu.edu.cn (X.S.); 17393181505@163.com (Y.Z.); van1131@163.com (J.F.); 2Key Laboratory of Yellow River Water Environment in Gansu Province, Lanzhou Jiaotong University, Lanzhou 730070, China

**Keywords:** TCE/water separation, surface modification, hydrophobicity-enhanced ZSM-5 zeolite, pervaporation

## Abstract

The hydrophobic nature of inorganic zeolite particles plays a crucial role in the efficacy of mixed matrix membranes (MMMs) for the separation of trichloroethylene (TCE) through pervaporation. This study presents a novel approach to further augment the hydrophobicity of ZSM-5. The ZSM-5 zeolite molecular sieve was subjected to modification using three different silane coupling agents, namely, n-octyltriethoxysilane (OTES), γ-methacryloxypropyltrimethoxysilane (KH-570), and γ-aminopropyltriethoxysilane (KH-550). The water contact angles of the resulting OTES@ZSM-5, KH-570@ZSM-5, and KH-550@ZSM-5 particles exhibited significant increases from 97.2° to 112.8°, 109.1°, and 102.7°, respectively, thereby indicating a notable enhancement in hydrophobicity. Subsequently, mixed matrix membranes (MMMs) were fabricated by incorporating the aforementioned silane-modified ZSM-5 particles into polydimethylsiloxane (PDMS), leading to a considerable improvement in the adsorption selectivity of these membranes towards trichloroethylene (TCE). The findings indicate that the PDMS membrane with a 20 wt.% OTES@ZSM-5 particle loading exhibits superior pervaporation performance. When subjected to a temperature of 30 °C, flow rate of 100 mL/min, and vacuum of 30 Kpa, the separation factor and total flux of a 3 × 10^−7^ wt.% TCE solution reach 328 and 155 gm^−2^·h^−1^, respectively. In comparison to the unmodified ZSM-5/PDMS membrane, the separation factor demonstrates a 41% increase, while the TCE flux experiences a 6% increase. Consequently, this approach effectively enhances the pervaporation separation capabilities of the PDMS membrane for TCE.

## 1. Introduction

The presence of volatile organic compounds (VOCs) in industrial wastewater is widely recognized as a significant global environmental concern due to the adverse effects these harmful compounds can cause [1,2]. Trichloroethylene (TCE), a prominent pollutant, exhibits carcinogenic, teratogenic, and mutagenic properties. Its ability to persist in air, water, and soil for extended periods poses a substantial risk to both ecological stability and human well-being. Various methods exist for the removal of TCE from water, including adsorption, extraction, and condensation techniques [3,4]. However, these approaches often necessitate complex regeneration procedures or involve phase transitions, adding to their complexity. The environment can be negatively impacted by the emission of harmful oxides resulting from combustion processes [5,6]. Organic/water systems can be effectively separated using pervaporation (PV). It has emerged as a promising method for separating TCE aqueous solutions due to its notable efficiency, minimal energy requirements, and absence of supplementary pollutants [7,8]. The selectivity of the pervaporation membrane, as per the solution diffusion model, is contingent upon the solubility and diffusion characteristics of the feed liquid within the membrane [9,10]. The affinity of the membrane material to the permeate, which refers to the solubility of the permeate in the membrane, is determined by its chemical properties. Consequently, the selection of appropriate membrane materials plays a crucial role in improving the pervaporation separation effect. 

Polydimethylsiloxane (PDMS), a hydrophobic rubbery polymer with a low glass transition temperature, possesses the ability to create a continuous and uninterrupted channel internally, facilitating the diffusion of macromolecular organic substances. As a result, PDMS has emerged as the prevailing choice for pervaporation membrane materials [11]. Nevertheless, the PDMS film exhibits a thin thickness, resulting in inadequate mechanical properties and notable flaws during practical implementation. In order to address this issue, the incorporation of hydrophobic particles into the PDMS polymer to form MMMs has emerged as a successful approach. Khan et al. [12] successfully prepared an MMM by using PDMS and ZIF-67 as polymers and fillers, respectively. A membrane loaded with 20 wt.% ZIF-67 showed a threefold increase in flux and a doubled separation coefficient for ethanol/water mixtures as compared with an unfilled PDMS membrane. Zhan et al. [13] conducted a study in which they synthesized ZHNTs and integrated them into PDMS to create a MMM for pervaporation of n-butanol. It was found that this MMM produced a separation factor of 61.3 at 40 °C for a 1 wt.% n-butanol aqueous solution, which was 58.7% higher than that of pure PDMS membrane. It is noteworthy that enhancing the hydrophobicity of particles can potentially enhance the adsorption affinity of MMM for organic matter, thereby further enhancing the pervaporation performance. Han et al. [14] conducted a study in which they sought to enhance the hydrophobicity of ZSM-5 particles through alkyl trichlorosilane modification and prepared a ZSM-5/PDMS MMM. The modified membranes, when compared to the unmodified membrane, exhibited increased separation factors and decreased total fluxes at the same zeolite loading.

ZSM-5, an zeolite molecular sieve, possesses a high silica/alumina ratio, an elevated surface energy, ample free volume, and a notable hydrophobicity. It exhibits a distinctive pore size distribution (5.4–5.6 A) and a well-organized crystal junction. Moreover, it demonstrates resistance to diverse solvents and high-temperature conditions, while displaying a remarkable adsorption selectivity for the TCE/water system [15,16]. Consequently, ZSM-5 has attracted attention as an effective doping particle for PDMS membranes, thus warranting further research. Cheng et al. [17] conducted an experiment in which they fabricated a composite membrane consisting of ZSM-5 zeolite and PDMS. A butanol solution was used to evaluate the pervaporation performance of this membrane. It was found that the incorporation of hydrophobic zeolite particles led to a decrease in water flux while maintaining the butanol flux, resulting in an increased separation coefficient for butanol. Specifically, when the butanol feed concentration was 1.5 wt.%, the PDMS MMM with 40 wt.% ZSM-5 addition exhibited a separation factor of 77 at a temperature of 47 °C. Additionally, the butanol flux and total flux were measured to be 62 gm^−2^·h^−1^ and 118 gm^−2^·h^−1^, respectively. Li et al. [18] conducted a study in which they utilized the dynamic negative pressure method to prepare a ZSM-5/PDMS/PVDF MMM for the recovery of phenol. The researchers found that when the ZSM-5 content was 40 wt.% and the coating time was 60 min, the separation coefficient and phenol permeation flux reached values of 4.56 and 5.78 gm^−2^·h^−1^, respectively. However, studies on enhancing the hydrophobicity of ZSM-5 are rare, compared to the numerous studies conducted on other conventional doped particles. ZSM-5’s chemical inertness may explain this lack of research.

In this work, we improved the pervaporation performance of PDMS-based composite membranes for TCE by augmenting the hydrophobicity of the integrated ZSM-5 particles. To achieve this, the ZSM-5 particles were subjected to modification using silane coupling agents OTES, KH-570, and KH-550. After adding modifiers, the anhydrous silylation occurred by hydrolysis of one or more of the oxyethyl groups [19], followed by a condensation reaction between the hydrolyzed modifiers and the surface hydroxyl groups of ZSM-5 particles to form the linkage of Si-O-Si, resulting in the attachment of silane group to the surface of ZSM-5 particles and resulting in the formation of OTES@ZSM-5, KH-570@ZSM-5, and KH-550@ZSM-5 particles, as illustrated in Figure 1. The present study aimed to investigate the morphology and hydrophobicity of silane-modified ZSM-5 particles using SEM, FTIR, XRD, and water contact angle measurement, respectively. Subsequently, these modified ZSM-5 particles were incorporated into PDMS to fabricate composite membranes, with the objective of enhancing the separation performance of TCE. Furthermore, the PV of pervaporation was also investigated in relation to preparation and operating conditions.

## 2. Experimental

### 2.1. Materials

Trichloroethylene (TCE) standard solution was purchased from Beijing Hengxin Ruihua Technology Co., Ltd., Beijing, China. N-N dimethylformamide (DMF) was purchased from Tianjin Damao Chemical Reagent Factory, Tianjin, China. N-heptane, tetraethyl orthosilicate (TEOS), dibutyltin dilaurate (DBDTL), and n-octyltriethoxysilane (OTES) were purchased from Fenlida Instruments Co., Ltd., Jiayuguan, China. γ-methacryloxypropyltrimethoxysilane (KH-570) and γ-aminopropyltriethoxysilane (KH-550) were purchased from Shanghai Macklin Biochemical Technology Co., Ltd., Shanghai, China. ZSM-5 zeolite (Si/Al = 300) was purchased from Nankai Catalyst Factory, Tianjing, China. Polydimethylsiloxane (PDMS) (Silicone Rubber 107, Mw5000) was purchased from Shandong Laizhou Jintai Silicon Industry Co., Ltd., Laizhou, China. Polyvinylidene fluoride (PVDF) was purchased from Shandong Xiya Chemical Co., Ltd., Linyi, China.

### 2.2. Hydrophobic Modification of ZSM-5 Zeolite

The ZSM-5 zeolite molecular sieve underwent hydrophobic modification using OTES, KH-570, and KH-550 silane coupling agents, resulting in the formation of modified zeolite particles OTES@ZSM-5, KH-570@ZSM-5, and KH-550@ZSM-5. Prior to utilization, ZSM-5 particles were calcined at 600 °C for 4 h in a muffle furnace. Subsequently, the particles were dried in an oven at 120 °C overnight to ensure complete removal of any template residues. The procedure for preparing OTES@ZSM-5 particles is as follows. ZSM-5 particles and OTES were dispersed in n-heptane and stirred for 8 h (W_ZSM-5_:W_OTES_:W_n-heptane_ = 3:1.5:50) [19]. Subsequently, the mixture was centrifuged and filtered, yielding a filter cake. The filter cake was then subjected to repeated washing with n-heptane to eliminate remaining OTES silane coupling agent. Following this, the filter cake was vacuum dried for 12 h at 80 °C. The preparation process for KH-570@ZSM-5 particles and KH-550@ZSM-5 particles is identical. The modifier (KH-550 or KH-570) was dissolved in n-heptane and stirred for a period of 2 min. In an 80 °C water bath pan, ZSM-5 particles were gradually added to the solution and maintained at that temperature for 10 h (W_ZSM-5_:W_modifier_:W_n-heptane_ = 10:1.5:50) [20,21]. The subsequent steps remain consistent with the aforementioned procedure.

### 2.3. Fabrication of MMMs

The preparation of PVDF carrier was performed as follows [22]. A solution containing a specific quantity of PVDF was prepared by dissolving it in DMF with a concentration of 11 wt.%. The solution was then subjected to stirring at room temperature for a duration of 8 h. A membrane maker was then used to flatten the solution onto a nonwoven fabric. Immediately following this step, the resulting membrane was immersed in water at 25 °C for a brief period of 2 s before being removed. The membrane was then dried in a fume hood for a duration of 30 min, resulting in the formation of an ultra-porous PVDF membrane [22]. The preparation process of a mixed casting solution containing ZSM-5 loaded PDMS involved dissolving PDMS and TEOS in n-heptane and stirring for 2 h. The modified zeolite particles were introduced and agitated at ambient temperature for a duration of 1 h. Subsequently, 0.5 g of catalyst DBDTL was incorporated, continuously agitated at ambient temperature for 30 min, and subsequently subjected to ultrasonic treatment for 30 min to mitigate agglomeration and eliminate foam from the solution. The proportions of polymer, solvent, crosslinking agent, and catalyst were established as W_PDMS_:W_n-heptane_:W_TEOS_:W_DBDTL_ = 30:70:2.5:0.5 (g) in accordance with the experimental design [23,24]. The preparation of the ZSM-5/PDMS/PVDF MMMs involved the utilization of the solution casting method [22]. Specifically, ZSM-5/PDMS mixed casting solution was applied for one minute to the PVDF substrate membrane’s external surface, after which it was transferred to a fume hood. Subsequently, the solvent was allowed to evaporate for 24 h at ambient temperature, followed by 12 h of curing in a vacuum oven at 120 °C. Ultimately, the modified ZSM-5/PDMS/PVDF MMMs were successfully prepared, as shown in Figure 2.

### 2.4. Characterization

To analyze the composite films’ morphology, high-resolution SEM images were obtained at an accelerating voltage of 2 kV using scanning electron microscopy (SEM, Gemini SEM 500, Carl Zeiss, Oberkochen, Germany). Infrared spectroscopy (FTIR, VERTEX 70, Brook, Karlsruhe, Germany) was performed in the range of 4000–400 cm^−1^ to analyze the changes in the chemical structures before and after crosslinking, such as the changes in functional groups. X-ray diffractometry (XRD, XRD-7000L, Brook, Karlsruhe, Germany) was used to characterize the composite films, and the changes in diffraction peak intensity and crystallinity were investigated. XRD analysis was performed with a film sample size of 20 mm × 20 mm, a scanning rate of 10 °·min^−1^, a scanning range of 5–60°, a Cu target, a ceramic light tube (X = 0.154 nm), a tube voltage 45 kV, and a tube current 200 mA. The changes in the hydrophilicity and hydrophobicity of the composite membranes were observed by a video optical contact angle measuring instrument (OCA25, Brook, Karlsruhe, Germany). An average value was determined for each segment by measuring three different points. The error range was ±0.6°.

### 2.5. Swelling Experiment

The composite membrane underwent a cutting process to obtain samples measuring 20 mm × 20 mm, which were subsequently subjected to vacuum drying for 12 h at 60 °C. These samples were then immersed in water and a 3 × 10^−7^ wt.% TCE aqueous solution for a period of 24 h each. The attainment of a constant mass for the samples indicates the achievement of swelling equilibrium in the film. The samples were removed at this point, and the liquid on the membrane surface was quickly wiped away with filter paper. Each individual sample was subjected to three measurements, and the resulting values were averaged. The swelling degree (SD) was determined using the following equation [25].
(1)$$SD=\frac{W_{s}-W_{D}}{W_{D}}\;$$
where $W_{D}$ is the dry weight of the sample, and $W_{s}$ is the wet weight of the sample.

### 2.6. PV Experiment

The PV apparatus is schematically illustrated in Figure 3. Our PV experiments were performed using a laboratory-scale membrane system with an effective membrane area of 2.46 × 10^−3^ m^2^ and a permeate pressure of 30 kpa. The feed solution between the feed tank and membrane cell was circulated at 100 mL/min. The vacuum pressure on the permeate side was controlled by a vacuum pump. When the operation reached a steady state, the sample is collected by liquid nitrogen condensation. A headspace chromatography–mass spectrometer (GC-MS 7000C, Agilent, Palo Alto, CA, USA) was used to analyze these samples quantitatively and qualitatively.

The permeation flux (J) and separation factor (α) were calculated as follows [26]:(2)$$J=M/\left(A\times t\right)\;$$
(3)$$\alpha =\left(y_{TCE}/y_{Water}\right)/(x_{TCE}/x_{Water})\;$$
where M is the total amount of permeate collected during the experimental time interval t of 1 h at steady state, A is the effective membrane area, and x and y represent the mole fractions of a component in the permeate and in the feed, respectively.

## 3. Results and Discussion

### 3.1. Characterization of Silane-Modified ZSM-5

The morphology of ZSM-5 particles before and after modification is depicted in Figure 4. All ZSM-5 zeolite particles demonstrate a consistent parallelepiped shape and a sleek surface, measuring approximately 1 × 0.5 × 0.2 μm^3^ in size [27]. This indicates that the modification layer is slender and does not impact the morphology of ZSM-5 particles.

The results of the FTIR spectrum are depicted in Figure 5, illustrating the presence of asymmetric tensile, symmetric tensile, and flexural vibrations of Si-O-Si at 1070, 800, and 450 cm^−1^, respectively [18]. The absorptions at 1250 cm^−1^ and 550 cm^−1^ corresponded to the presence of five-membered ring chains and different ring structures, respectively [19]. The characteristic peaks at 2960 cm^−1^ and 2870 cm^−1^ correspond to the asymmetric tensile vibration of -CH_3_, which originates from the methyl groups present in the silane coupling agent [20]. This indirectly proved that the silane coupling agent was successfully grafted on the surface of ZSM-5 particles. The -OH stretching of the modified ZSM-5 at 3450 cm^−1^ is comparatively weakened when compared to the original peak. Furthermore, the modified ZSM-5 exhibits significant enhancements in the bands at 1070, 800, and 450 cm^−1^. The observed alteration indicated the emergence of novel Si-O-Si bonds within ZSM-5 zeolite, thereby confirming the successful grafting of OTES, KH-570, and KH-550 onto the surface of ZSM-5 particles [19].

The XRD findings, as depicted in Figure 6, exhibited that the ZSM-5 zeolite modified with silane possessed five diffraction peaks at approximately 8°, 8.9°, 23.1°, 23.3°, and 24°, which corresponded to the distinctive peaks of MFI zeolite topology [28]. These peaks were consistent with the reflection peaks of unmodified ZSM-5, signifying the preservation of the crystal structure of ZSM-5 following silane modification.

Figure 7 shows the hydrophobicity of modified ZSM-5 particles. The unmodified ZSM-5 exhibited a water contact angle of 97.2°, thereby confirming its hydrophobic nature. Conversely, the OTES@ZSM-5, KH-570@ZSM-5, and KH-550@ZSM-5, which were modified with OTES, KH-570, and KH-550, respectively, displayed water contact angles of 112.8°, 109.1°, and 102.7°. ZSM-5 exhibits a significant increase in water contact angle as a result of silane modification. Furthermore, the hydrophobicity of OTES@ZSM-5 surpasses that of KH-570@ZSM-5 and KH-550@ZSM-5, a distinction attributable to the lower polarity of OTES compared to KH-570 and KH-550. This polarity difference arises from the longer alkyl chain present in OTES, as it is well established that an increase in alkyl chain length corresponds to an augmented hydrophobicity [29,30].

### 3.2. Characterization and PV Performance of the MMMs

Based on the findings presented above, silane modification enhances the hydrophobic properties of ZSM-5 without adversely affecting its chemical or crystal structure. Consequently, MMMs were fabricated by incorporating modified ZSM-5 particles into PDMS polymer, and an investigation was conducted on the morphology, water contact angle, swelling degree, and PV performance of the MMMs. 

Figure 8 illustrates the SEM cross-section of ZSM-5/PDMS MMMs prior to and following modification, with a particle loading of 20 wt.%. Both the unmodified and modified ZSM-5/PDMS MMMs exhibited comparable morphology and thickness. The PVDF support surface was coated with all PDMS polymer layers. The resulting composite membrane exhibited a dense structure with no discernible defects. Notably, the zeolite particles in the OTES-modified ZSM-5/PDMS MMM displayed a homogeneous dispersion, as depicted in Figure 8b. The introduced ZSM-5 was effectively embedded within the PDMS phase, without any noticeable aggregation, unlike other modified MMMs where some ZSM-5 particles tended to agglomerate. As a result of the long n-octyl chain on the surface of OTES@ZSM-5 particles entangling with the PDMS chain, the zeolite particles interact more strongly with the polymer [19,31].

Figure 9 shows the water contact angles of different modified MMMs. The water contact angle of the unmodified ZSM-5/PDMS mixed matrix membrane (MMM) was measured to be 123.4°. In contrast, the water contact angles of the ZSM-5/PDMS MMMs modified with OTES, KH-570, and KH-550 were found to be 130.5°, 127.8°, and 124.2°, respectively. Therefore, the hydrophobicity of the PDMS membrane was enhanced after incorporating silane-modified ZSM-5 particles. Interestingly, the water contact angle for the modified ZSM-5/PDMS MMMs was found to be nearly identical to that of the unmodified MMM. This can be attributed to the fact that the polymer encapsulates the zeolite, and the observed improvement in hydrophobicity is primarily due to the increase in surface roughness [17].

The membranes that were prepared underwent immersion in water and a 3 × 10^−7^ wt.% TCE solution for the purpose of conducting swelling experiments. The findings of these experiments are depicted in Figure 10. The observed swelling degree can be ranked in the following order: OTES@ZSM-5/PDMS > KH-570@ZSM-5/PDMS > KH-550@ZSM-5/PDMS > unmodified ZSM-5/PDMS. The hydrophobic nature of the ZSM-5 particles, both before and after modification, may explain this trend [32]. The ZSM-5 particles possess a porous structure that offers a greater volume for the molecules within the membrane material. Additionally, the silane-modified ZSM-5 particles have enhanced hydrophobicity, resulting in increased affinity of MMM for trichloroethylene (TCE) [33]. The silane-modified ZSM-5 doped MMMs demonstrate improved selective adsorption for TCE, as evidenced by their consistently low swelling degree in water.

The PV performance of ZSM-5@PDMS MMMs was examined both before and after modification. This examination took place under specific conditions, including a feed TCE concentration of 3 × 10^−7^ wt.%, a feed temperature and flow rate 0f 30 °C and 100 mL/min, and vacuum degree of 30 Kpa. As shown in Figure 11, the PV performance of all modified ZSM-5/PDMS MMMs was significantly improved (Figure 11a). Doped ZSM-5 particles are more hydrophobic, which can explain this enhancement [33]. In comparison to unmodified MMMs, MMMs incorporating modified ZSM-5 particles exhibited a greater separation factor and marginally reduced total flux, potentially attributed to the heightened hydrophobic nature of the modified ZSM-5 particles. The inclusion of modified ZSM-5 particles in MMMs augmented the affinity for TCE and impeded the mass transfer of water [34]. Consequently, the TCE flux of MMMs containing modified ZSM-5 particles exhibited a gradual increase, while the water flux was notably lower than that of MMMs containing unmodified ZSM-5 particles (Figure 11b). Simultaneously, it is observed that the OTES@ZSM-5/PDMS mixed matrix membrane (MMM) exhibiting the highest hydrophobicity also possesses the highest separation coefficient, albeit with the lowest total flux compared to other MMMs. It is noteworthy that the OTES@ZSM-5/PDMS MMM, when compared to the unmodified ZSM-5/PDMS MMM, demonstrates a 41% increase in separation factor and a 6% increase in TCE flux when loaded with 20 wt.%. Therefore, enhancing ZSM-5’s hydrophobicity is an effective approach for improving the permeation and separation performance of PDMS membranes for TCE.

### 3.3. Effect of Zeolite Loading on OTES@ZSM-5/PDMS MMM

The effect of particle loading on MMMs was investigated, and the PDMS composite membrane doped with OTES@ZSM-5 molecular sieve demonstrated the highest separation factor. Figure 12 displays the SEM cross-section of OTES@ZSM-5/PDMS MMMs with different zeolite loadings. When the loading of OTES@ZSM-5 particles is 10 wt.% (Figure 12b), the composite membrane’s surface gradually exhibited roughness, and the distribution of doped particles became sparse. When the loading amount of OTES@ZSM-5 reached 20 wt.% (Figure 12c), the doped particles dispersed uniformly in the PDMS matrix, suggesting that the membrane material is compatible with the particles [35]. However, as the loading of OTES@ZSM-5 exceeded 30 wt.% (Figure 12d–e), aggregation of OTES@ZSM-5 particles occurred on the surface of the PDMS membrane, accompanied by the appearance of local defects. This observation indicates that the saturation point for OTES@ZSM-5 loading in PDMS is approximately 20 wt.%.

The water contact angles of OTES@ZSM-5/PDMS MMMs with varying loadings were measured, and the corresponding results are presented in Figure 13. It can be seen that the water contact angle of the composite film increased proportionally with the increase in particle loading. Notably, when the particle loading reached 40 wt.%, the water contact angle attained its maximum value of 144.2°. According to these findings, higher particle loading promotes composite membrane hydrophobicity. This is because the incorporation of zeolite enhances the surface roughness of the composite membrane and augments its resistance to wetting [35].

The pervaporation performance of OTES@ZSM-5/PDMS MMMs was examined to determine the impact of zeolite particle loading. The investigation was conducted under specific conditions, including a feed TCE concentration of 3 × 10^−7^ wt.%, a feed temperature and flow rate of 30 °C and 100 mL/min, and vacuum degree of 30 Kpa. Figure 14 illustrates the results. When the particle loading of OTES@ZSM-5 increased, the separation factor of MMMs increased significantly. After reaching its peak at 20 wt.%, the separation factor began to decline. It is noteworthy that the total flux and separation factor exhibit contrasting trends. This phenomenon can be comprehended as follows: as the loading of OTES@ZSM-5 particles rises from 0 wt.% to 20 wt.%, the composite membrane’s high hydrophobicity reduces its ability to selectively interact with water molecules. Simultaneously, it strengthens the affinity between TCE molecules and the composite membrane, leading to a significant decline in water flux and a rapid increase in TCE flux (Figure 14b). Consequently, the total flux diminishes while TCE selectivity increases. When the loading of OTES@ZSM-5 particles exceeds 20 wt.%, defects are more prone to occur in mixed matrix membranes (MMMs) due to the ongoing increase in particle loading [36]. In addition to voids formed by particle aggregation, non-selective microvoids may occur between particles and membranes. These defects can create additional channels for mass transfer, as water molecules are smaller than TCE molecules, facilitating the diffusion of water molecules and leading to a sudden increase in water flux (Figure 14b). As a result, the separation factor gradually decreases. The findings indicate that the OTES@ZSM-5/PDMS mixed matrix membrane (MMM) containing 20 wt.% zeolite loading exhibited superior pervaporation separation performance when applied to a 3 × 10^−7^ wt.% TCE aqueous solution. Notably, the separation factor and TCE flux achieved values of 278 and 0.928 gm^−2^·h^−1^, respectively.

### 3.4. Effect of PVDF Content on OTES@ZSM-5/PDMS MMM

The pervaporation performance of the composite membrane was investigated under specific conditions, including a feed TCE concentration of 3 × 10^−7^ wt.%, a feed temperature and flow rate of 30 °C and 100 mL/min, and vacuum degree of 30 Kpa. The effect of PVDF content in the membrane carrier on the pervaporation performance was analyzed, as depicted in Figure 15. With increasing PVDF content, the separation factor initially increases and then continuously decreases, reaching its maximum value at 11 wt.%. Meanwhile, the total flux exhibits a declining trend. The explanation for this phenomenon can be elucidated as follows: at a PVDF content of 10 wt.%, the membrane carrier exhibits a thin structure with a particle arrangement on its surface, leading to a relatively rough texture. Consequently, the hydrophobicity of the membrane is enhanced, resulting in an increased affinity for TCE (as shown in Figure 15b) [37,38]. Consequently, the overall flux gradually diminishes while the selectivity increases. On the other hand, when the PVDF content surpasses 11 wt.%, the membrane carrier gradually transforms into a denser and smoother structure, thereby reducing its roughness. In contrast, the thickness of the membrane exhibits an increase, consequently resulting in an increase in the mass transfer resistance of both water molecules and TCE molecules. Furthermore, a higher concentration of PVDF leads to a greater presence of polymer chains. Thus, the membrane layer becomes thicker and solution viscosity increases. During the preparation of the substrate membrane, a larger quantity of polymers is retained on the membrane surface, forming a barrier that ultimately leads to a reduction in both water flux and TCE flux (Figure 15b).

### 3.5. Effect of Feed Concentration on PV Performance

The pervaporation performance of OTES@ZSM-5/PDMS MMM was examined to assess the impact of varying feed TCE concentrations (ranging from 1 × 10^−7^ wt.% to 5 × 10^−7^ wt.%) under specific conditions, including a feed temperature and flow rate of 30 °C and 100 mL/min, and vacuum degree of 30 Kpa. Figure 16 illustrates the findings. The total flux exhibited a gradual increase, whereas the separation factor demonstrated a decrease as the TCE concentration increased. This observation aligns with previous literature findings [14,39,40]. As TCE content in the feed solution increases, the swelling degree of the composite membrane increases due to its strong affinity for TCE. This, in turn, facilitates the diffusion of TCE molecules and water molecules, resulting in a gradual increase in total flux. Permeation selectivity is determined by the dissolution and diffusion processes of the penetrating molecules within the composite membrane [41,42]. The composite membrane’s hydrophobic nature results in a greater affinity of TCE molecules to the membrane compared to water molecules. Nevertheless, during the diffusion separation process, water exhibits a significantly higher diffusion rate than TCE due to its smaller molecular size. Consequently, the impact of diffusivity on pervaporation outweighs that of solubility in this scenario. Consequently, an increase in TCE content yields a higher total flux and a lower separation factor.

### 3.6. Effect of Feed Temperature on PV Performance 

The impact of varying feed temperatures on the pervaporation efficiency of OTES@ZSM-5/PDMS MMMs was examined. This investigation was conducted while maintaining a feed trichloroethylene (TCE) concentration of 3 × 10^−7^ wt.%, a vacuum degree of 30 Kpa, and a feed flow rate of 100 mL/min, as depicted in Figure 17. According to the figure, total flux, water flux, and TCE flux all increased as temperature increased. This phenomenon can be attributed to the direct relationship between temperature and vapor pressure, where an increase in temperature leads to an increase in the vapor pressure of the components present in the feed solution upstream of the membrane [43]. Consequently, this elevation in temperature also enhances the mass transfer driving force across the membrane. Furthermore, the heightened temperature promotes the activity of the polymer chain segment, consequently augmenting the free volume between molecules. As a result, this acceleration in mass transfer facilitates the permeation of components from the upstream to the downstream of the membrane [44,45]. It is noteworthy that the separation factor exhibits a consistent upward trend. With increasing temperature, the permeate’s saturated vapor pressure rises, surpassing that of water. Consequently, the growth rate of TCE flux becomes more pronounced compared to that of water flux, resulting in an augmented separation factor [46].

### 3.7. Effect of Feed Flow Rate on PV Performance

The present study examined the impact of varying feed flow rates on the PV performance of OTES@ZSM-5/PDMS MMM. This investigation was conducted under specific conditions, including a feed TCE concentration of 3 × 10^−7^ wt.%, a feed temperature of 30 °C, and vacuum degree of 30 Kpa. Figure 18 shows that as the liquid flow rate increases, the total flux exhibits a gradual increase, albeit with a diminishing rate of growth. This phenomenon can be attributed to the accelerated flow velocity, resulting in a thinner boundary layer, a reduced concentration polarization, a decreased liquid mass transfer resistance, and ultimately an enhanced total flux [47,48]. When the flow rate reaches a certain threshold, the enhancement of mass transfer diminishes, leading to a gradual deceleration in the increase of flux. Furthermore, the impact of flow rate on the separation coefficient remains relatively stable. This is because as the feed flow rate rises, both water flux and TCE flux experience a gradual increase, yet the relative changes between the two are minimal, resulting in negligible influence of flow rate variations on the separation coefficient [49,50].

### 3.8. Effect of Vacuum Degree on PV Performance

The pervaporation performance of OTES@ZSM-5/PDMS MMM was investigated under specific conditions, including a feed TCE concentration of 3 × 10^−7^ wt.% and a feed temperature and flow rate of 30 °C and 100 mL/min. The impact of vacuum degree on the pervaporation performance was analyzed, as depicted in Figure 19. It is evident from Figure 19a that an increase in vacuum on the permeation side from 15 Kpa to 35 Kpa results in an upward trend in the total flux. This is due to the enhanced driving force of water and TCE molecules through the membrane, leading to increased water and TCE flux (Figure 19b) and subsequently higher overall total flux [51,52]. Furthermore, it was found that the separation factor exhibits an upward trend as the vacuum degree on the permeation side increases. Because under higher vacuum conditions, the volatilization of TCE molecules on the permeate side of the membrane becomes significantly more facile compared to that of water molecules. Consequently, this results in a more rapid diffusion of TCE molecules relative to water molecules, ultimately enhancing the selectivity and robustness of the membrane [24].

## 4. Conclusions

This paper presents a novel approach aimed at enhancing the hydrophobicity of ZSM-5 particles and improving the pervaporation efficiency of mixed matrix membranes (MMMs) for the separation of trichloroethylene (TCE). The surface of ZSM-5 particles was successfully modified through the grafting of OTES, KH-570, and KH-550, resulting in a significant enhancement of their hydrophobic properties. The prepared MMMs exhibited superior adsorption selectivity and pervaporation performance for TCE, as evidenced by the separation factor and total flux achieved by the OTES@ZSM-5/PDMS MMM. Specifically, at operating conditions of 55 °C, a flow rate of 100 mL/min, and a vacuum degree of 30 Kpa, the separation factor and total flux for a 3 × 10^−7^ wt.% TCE aqueous solution reached 328 and 155 gm^−2^·h^−1^, respectively. The utilization of the OTES@ZSM-5/PDMS membrane in this study resulted in a 41% increase in separation factor and a 6% increase in TCE flux compared to the unmodified ZSM-5/PDMS membrane. Additionally, Table 1 provides a comprehensive overview of previously reported membranes for TCE pervaporation separation, highlighting the OTES@ZSM-5/PDMS membrane as the most superior in terms of separation factor. Consequently, the hydrophobicity enhancement approach presented in this research effectively enhances the performance of MMM pervaporation for TCE aqueous solutions.

## Figures and Tables

**Figure 1 polymers-15-03777-f001:**
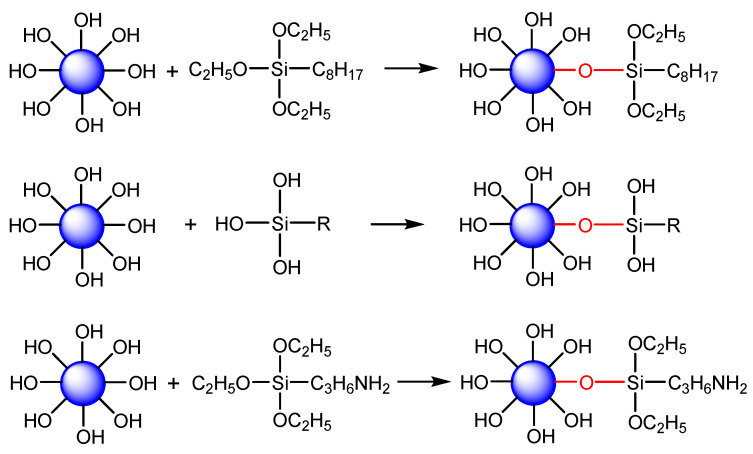
Schematic of ZSM-5 zeolite’s hydrophobic modification.

**Figure 2 polymers-15-03777-f002:**
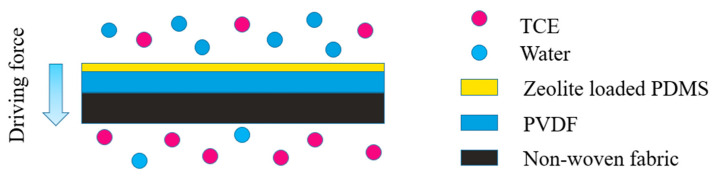
Schematic diagram of structure of ZSM-5/PDMS/PVDF MMMs.

**Figure 3 polymers-15-03777-f003:**
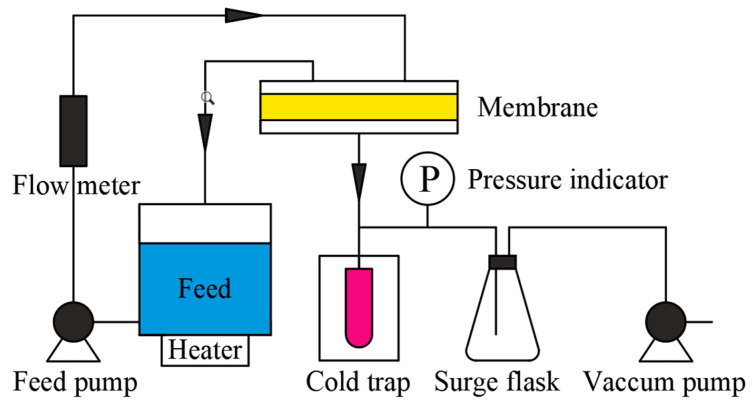
Schematic diagram of pervaporation apparatus.

**Figure 4 polymers-15-03777-f004:**
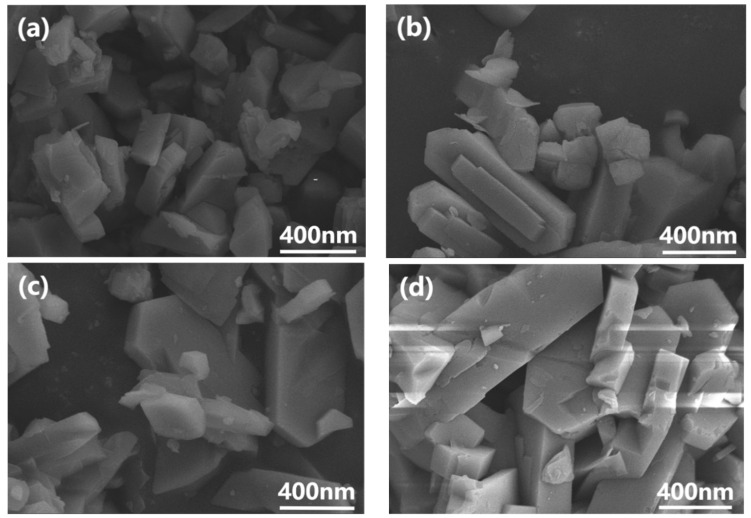
SEM images of (**a**) unmodified ZSM-5, (**b**) OTES@ZSM-5, (**c**) KH-570@ZSM-5, and (**d**) KH-550@ZSM-5.

**Figure 5 polymers-15-03777-f005:**
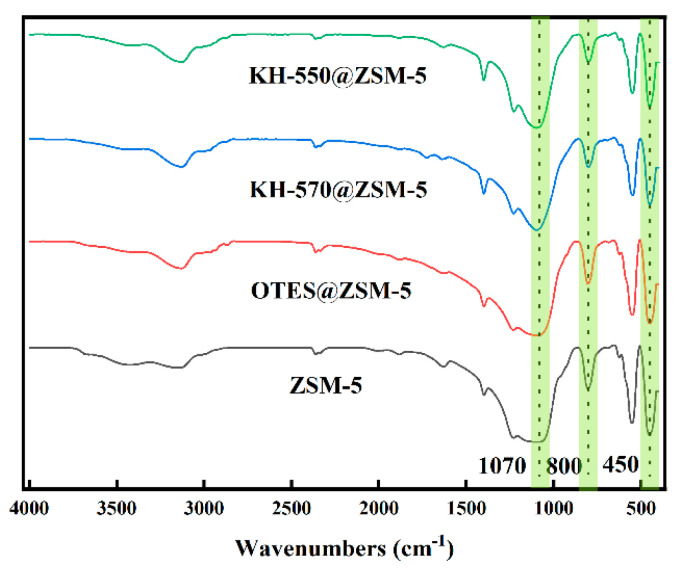
FTIR spectra of unmodified ZSM-5, OTES@ZSM-5, KH-570@ZSM-5, and KH-550@ZSM-5.

**Figure 6 polymers-15-03777-f006:**
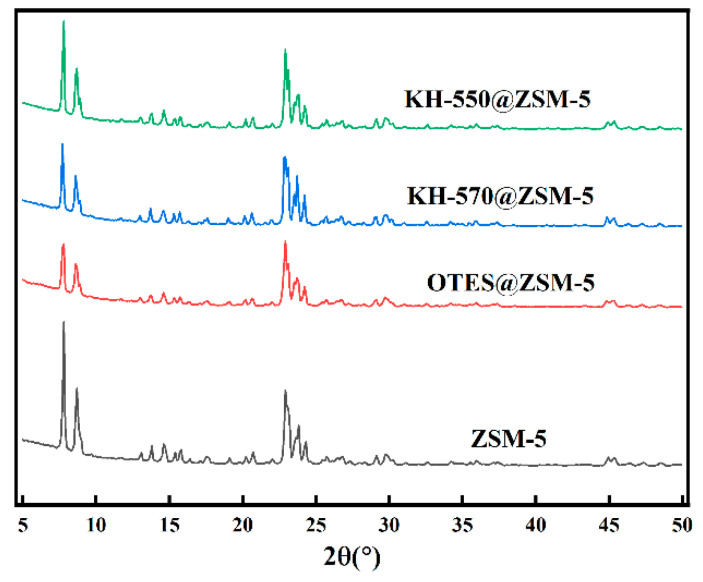
XRD patterns of unmodified ZSM-5, OTES@ZSM-5, KH-570@ZSM-5, and KH-550@ZSM-5.

**Figure 7 polymers-15-03777-f007:**
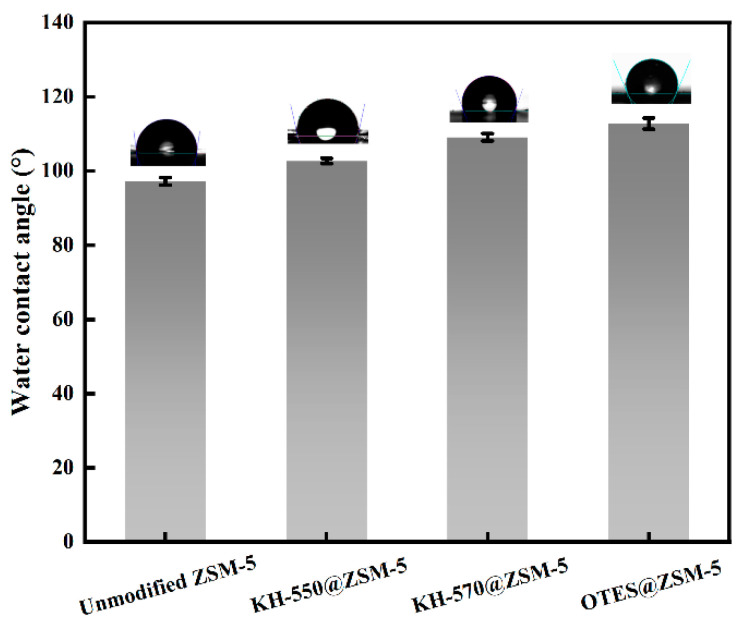
Water contact angles of unmodified ZSM-5, OTES@ZSM-5, KH-570@ZSM-5, and KH-550@ZSM-5.

**Figure 8 polymers-15-03777-f008:**
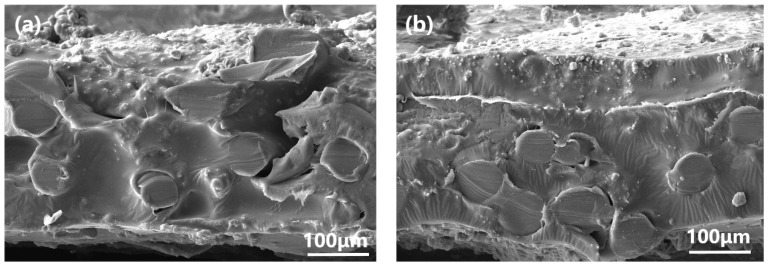
Cross-section images of the MMMs doped with 20 wt.% (**a**) unmodified ZSM-5, (**b**) OTES@ZSM-5, (**c**) KH-570@ZSM-5, and (**d**) KH-550@ZSM-5 particles.

**Figure 9 polymers-15-03777-f009:**
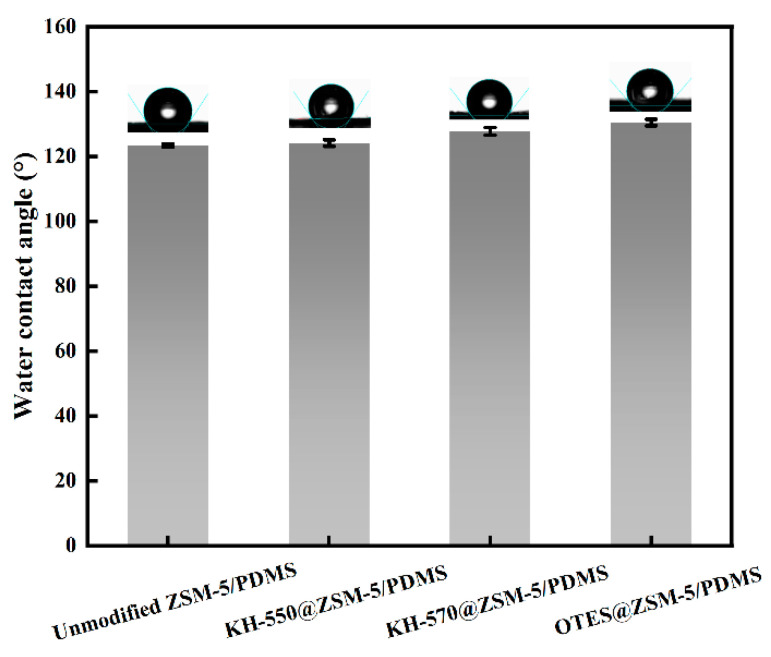
Water contact angles of the MMMs doped with 20 wt.% unmodified ZSM-5, KH-550@ZSM-5, KH-570@ZSM-5, and OTES@ZSM-5 particles.

**Figure 10 polymers-15-03777-f010:**
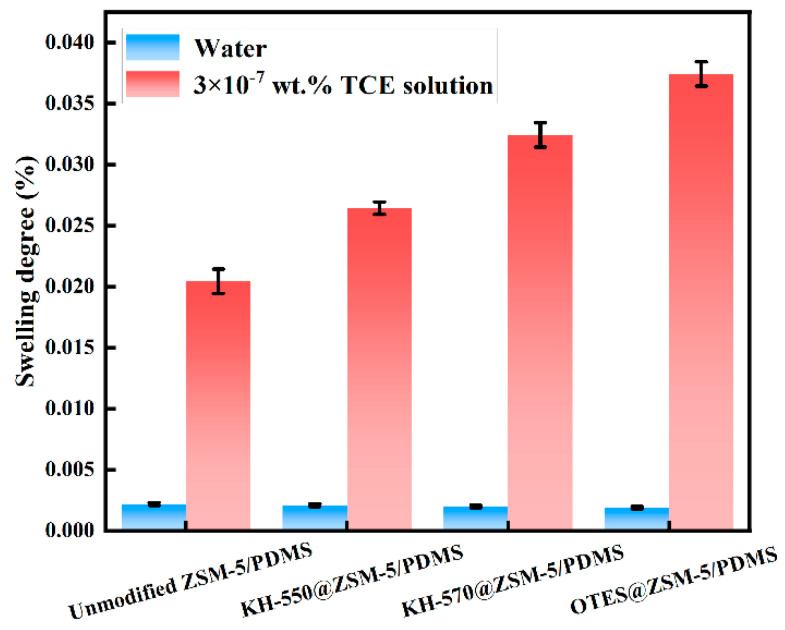
Swelling degrees of the MMMs doped with unmodified ZSM-5, KH-550@ZSM-5, KH-570@ZSM-5, and OTES@ZSM-5 particles.

**Figure 11 polymers-15-03777-f011:**
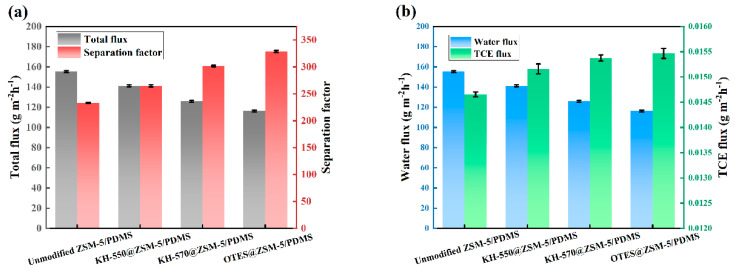
PV performances of the MMMs doped with unmodified ZSM-5, KH-550@ZSM-5, KH-570@ZSM-5, and OTES@ZSM-5 particles in a 3 × 10^−7^ wt.% TCE solution at 30 °C. (**a**) Effects on total flux and separation factors, (**b**) effect on water flux and TCE flux.

**Figure 12 polymers-15-03777-f012:**
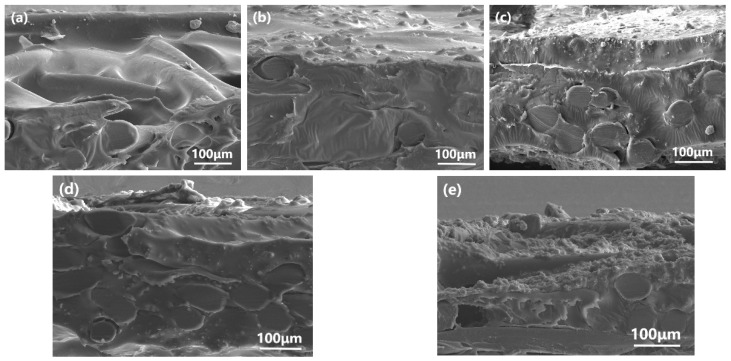
Cross-section images of OTES@ZSM-5/PDMS membranes with different particle loading amounts: (**a**) 0 wt.%, (**b**) 10 wt.%, (**c**) 20 wt.%, (**d**) 30 wt.%, and (**e**) 40 wt.%.

**Figure 13 polymers-15-03777-f013:**
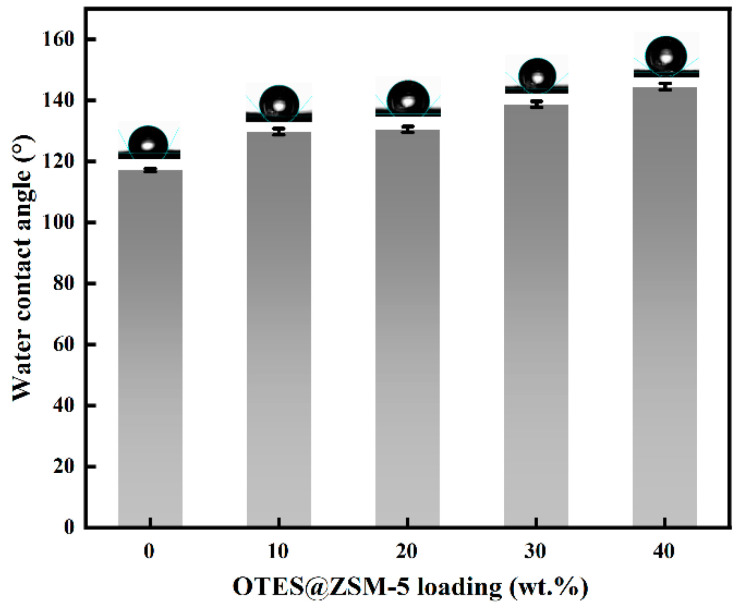
Water contact angles of OTES@ZSM-5/PDMS membranes with different particle loading amounts.

**Figure 14 polymers-15-03777-f014:**
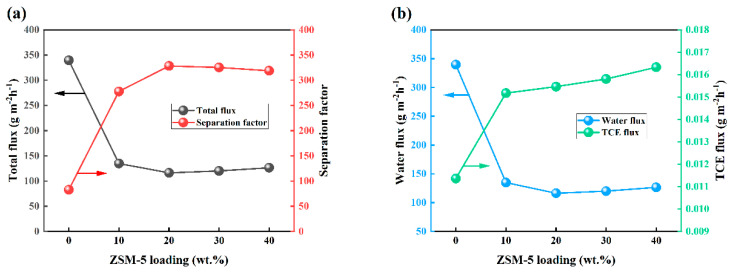
PV performance of OTES@ZSM-5/PDMS membranes with different particle loading amounts. (**a**) Effects on total flux and separation factors, (**b**) effect on water flux and TCE flux.

**Figure 15 polymers-15-03777-f015:**
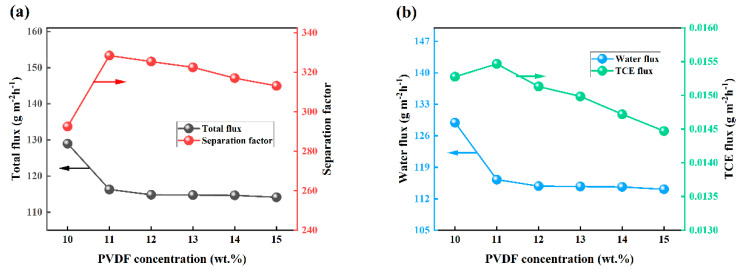
PV performance of OTES@ZSM-5/PDMS membranes with different PVDF concentrations. (**a**) Effects on total flux and separation factors, (**b**) effect on water flux and TCE flux.

**Figure 16 polymers-15-03777-f016:**
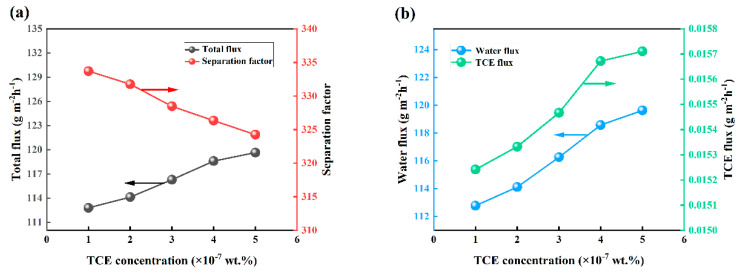
Effect of feed concentration on PV performance. (**a**) Effects on total flux and separation factors, (**b**) effect on water flux and TCE flux.

**Figure 17 polymers-15-03777-f017:**
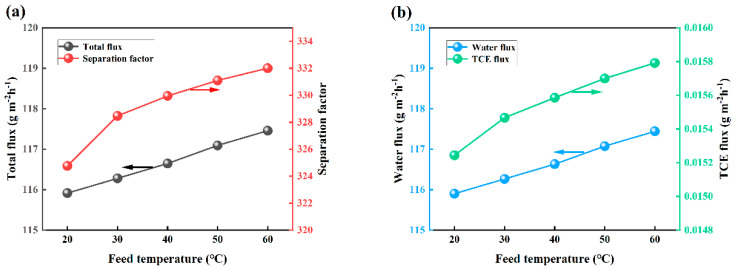
Effect of feed temperature on PV performance. (**a**) Effects on total flux and separation factors, (**b**) effect on water flux and TCE flux.

**Figure 18 polymers-15-03777-f018:**
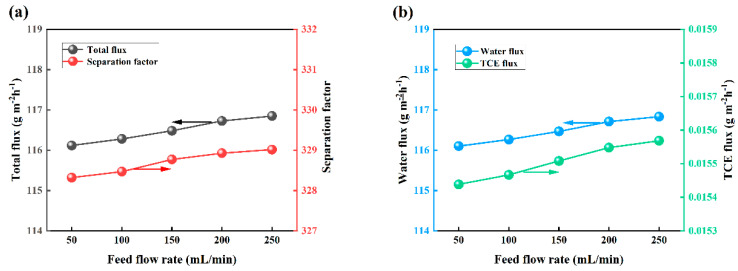
Effect of feed flow rate on PV performance. (**a**) Effects on total flux and separation factors, (**b**) effect on water flux and TCE flux.

**Figure 19 polymers-15-03777-f019:**
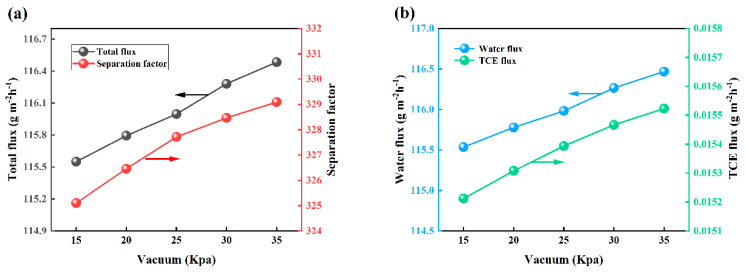
Effect of vacuum degree on PV performance. (**a**) Effects on total flux and separation factors, (**b**) effect on water flux and TCE flux.

**Table 1 polymers-15-03777-t001:** PV performance of TCE separated by different membranes.

Membrane	Feed Concentration (wt.%)	Feed Temperature (℃)	Particle Loading (wt.%)	Separation Factor	Total Flux (g m−2 h−1)	Reference
silicalite-1	1 × 10^−4^	30	-	10	4	[42]
polyvinyl acetate	4.57 × 10^−4^	25	-	110	310	[45]
poly(acrylate-co-acrylic)	2.01 × 10^−1^	25	-	108	638	[50]
PDMS	2.4 × 10^−4^	55	-	104	400	[52]
PDMS hollow fiber	9 × 10^−4^	20	-	140	190	[38]
OTES@ZSM-5/PDMS	3 × 10^−7^	30	20	328	155	This work

## Data Availability

All relevant data used in this study can be obtained from the corresponding authors upon reasonable request.

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
