# Peer review of "Improving the Pervaporation Performance of PDMS Membranes for Trichloroethylene by Incorporating Silane-Modified ZSM-5 Zeolite"

_polymers, 2023, doi:10.3390/polym15183777_

Round 1

Reviewer 1 Report

The manuscript "Improving the pervaporation performance of PDMS membranes for trichloroethylene by incorporating silane-modified ZSM-5 zeolite" reports on the hydrophobization of ZSM-5 particles to produce new composites membranes used in pervaporation of TCE. The overall composition is well structured and the results support the conclusions. I have only minor observations regarding this paper:

- Figure 1- usually these reactions lead to mixed products, by hydrolysis of alkoxysilane, the newly formed bridges must be emphasized;

-section 2.3.- A schematic representation of MMMs fabrication process must be added;

-section 2.3. line 147-the quantity of DBTL must be precisely mentioned;

-section 2.3.-a reference must be added at lines 152-154;

-section 2.5. why authors chose the concentration of TCE of 3x10-7 wt.%?

-section 3.1.- a mapping of the elements to highlight the presence of Si by EDX must be added at Figure 3;

-the comments regarding the FTIR characterisation at lines 214-217 are made by assumptions. The spectra were normalized? Usually, a ratio between two significant bands are used to highlight the changes. The newly formed bands are Si-O-Metal and Si-O-Si, only some of these bands are mentioned. Please revise.

-in Figures 14-18- error bars must be added.

-section 4- the reference to Table 1 at lines 481-485 must be included in section 3, here only a concluding remark.

Based on these remarks I suggest the publication of this article after Minor revision.

Reviewer 2 Report

The manuscript is generally clear and the discussion on the results seems convincing. I suggest acceptance with minor revisions:

-The sentence “The environment can be negatively impacted by the emission of harmful
oxides resulting from chemical precipitation and combustion processes” is not obvious in term of “oxides form precipitation”.

-In Figure 1, why only one SiOR group is expected to react?

-Concerning the sentence "The filter cake was then subjected to repeated washing with n-heptane to eliminate any remaining coupling agent." These latter are not mentioned, what are them?

-A scheme showing the preparation of ZSM-5/PDMS/PVDF MMMs might be useful

-The recycling of the membrane should be presented/discussed, highlighting stability of morphology and performance after use.
